# DynamicBEV: Leveraging Dynamic Queries and Temporal Context for 3D Object Detection

## Abstract

3D object detection is crucial for applications like autonomous driving and robotics. While query-based 3D object detection for BEV (Bird's Eye View) images has seen significant advancements, most existing methods follows the paradigm of static query. Such paradigm is incapable of adapting to complex spatial-temporal relationships in the scene. To solve this problem, we introduce a new paradigm in DynamicBEV, a novel approach that employs dynamic queries for BEV-based 3D object detection. In contrast to static queries, the proposed dynamic queries exploit K-means clustering and Top-K Attention in a creative way to aggregate information more effectively from both local and distant feature, which enable DynamicBEV to adapt iteratively to complex scenes. To further boost efficiency, DynamicBEV incorporates a Lightweight Temporal Fusion Module (LTFM), designed for efficient temporal context integration with a significant computation reduction. Additionally, a custom-designed Diversity Loss ensures a balanced feature representation across scenarios. Extensive experiments on the nuScenes dataset validate the effectiveness of DynamicBEV, establishing a new state-of-the-art and heralding a paradigm-level breakthrough in query-based BEV object detection.

## 1 Introduction

3D object detection is a pivotal task in various applications like autonomous driving, robotics, and surveillance Huang et al. (2021); Huang & Huang (2022a); Li et al. (2022c;b); Park et al. (2022). In the field of 3D object detection, BEV (Bird's Eye View) algorithms have achieved increasing prominence due to their ability to provide a top-down perspective, simplifying complex 3D scenes into 2D representations. This perspective aids in reducing computational complexity and enhancing the clarity of object localization. However, traditional query-based BEV methods have mainly exploited static queries Wang et al. (2022); Liu et al. (2022a;b), the query weights are learned from the training phase and keep fixed during inference. This static nature limits the model's ability to leverage both spatial and temporal context effectively and adapt to complex scenes. We argue that evolving from static to dynamic queries can initiate a new paradigm of 3D object detection, which will exploit more robust mechanisms to adaptively capture complex spatial-temporal relationships. Figure 2 presents static query-based methods, such as DETR3D Wang et al. (2022), employ queries that are learnable during training but remain fixed during inference. In contrast, our dynamic query-based method, DynamicBEV, allows for queries to adapt to the input data in an iterative way, offering greater generalization and flexibility.

In this vein, we introduce DynamicBEV, a novel method that pioneers dynamic queries in query-based 3D object detection. Unlike traditional static queries in BEV-based methods, the proposed dynamic queries are subject to iterative adapt in complex scenes. Specifically, we exploit feature clustering to generate adaptive scene representation, and develop a Top-K Attention mechanism where the query adapts to most relevant top-k clusters. This dynamism allows each query to aggregate information adaptively from both local and distant feature clusters, thereby significantly enhancing the model's ability to capture complex 3D scenarios.

Along with Top-K Attention scheme, we introduce a Diversity Loss that balances the attention weights to ensure that not only the most relevant but also the less prominent features are considered. This not only elevates the detection accuracy but also boosts the model's robustness and adaptability to different scenarios.

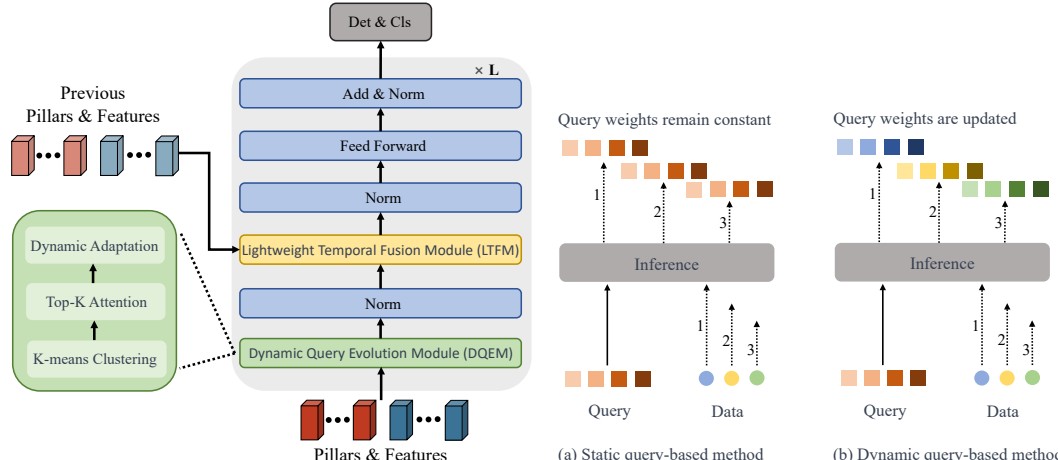

Figure 1: The architecture of DynamicBEV. The process starts with the extraction of features using a backbone network. Then, the features are clustered around each query through K-means clustering. Next, Top-K Attention Aggregation is applied to adaptively update each query. Finally, the updated queries are used for 3D object prediction.

Figure 2: Comparison between static query-based and dynamic query-based methods.

To further improve the efficiency of DynamicBEV, we propose Lightweight Temporal Fusion Module (LTFM). Contrary to traditional temporal fusion approaches that brings a significant computational burden, LTFM reuses the already computed dynamic queries and their associated feature clusters, which gets rid of the heavy cost of the specific feature extraction in traditional temporal fusion approaches and significantly boosts the efficiency the of temporal context incorporation.

We rigorously evaluate DynamicBEV on the nuScenes dataset, where it shows a significant gap over the state-of-the-art methods in terms of both accuracy and efficiency.

## 2 RELATED WORK

### 2.1 QUERY-BASED OBJECT DETECTION IN 2D AND 3D

Query-based object detection has gained significant advancements thanks to the introduction of the Transformer architecture Vaswani et al. (2017). Primary works like DETR Carion et al. (2020) adopted a static query-based approach where queries are used to represent potential objects but do not adapt during the detection process. Various works Zhu et al. (2020); Sun et al. (2021); Gao et al. (2022) have focused on accelerating the convergence or improving the efficiency of these static query-based methods. However, these models, even when extended to 3D space Wang et al. (2022); Liu et al. (2022a), inherently lack the ability to adapt queries to complex spatial and temporal relationships within the data. Our work diverges from this static paradigm by introducing dynamic queries that iteratively adapt during detection, effectively constituting a new paradigm in query-based object detection.

### 2.2 MONOCULAR AND MULTIVIEW 3D OBJECT DETECTION

Monocular 3D object detection Wang et al. (2019); Reading et al. (2021); Wang et al. (2021) and multiview approaches Philion & Fidler (2020); Huang et al. (2021) have been widely studied for generating 3D bounding boxes from 2D images. While effective, these methods generally operate under a static framework where features are extracted and used without further adaptation. Our work, DynamicBEV, enhances this by dynamically adapting the queries in BEV space to capture both local and distant relationships, thus presenting a novel approach in the realm of 3D object detection.

### 2.3 STATIC VS. DYNAMIC PARADIGMS IN BEV OBJECT DETECTION

BEV-based object detection has seen various advancements Pan et al. (2020); Philion & Fidler (2020); Huang et al. (2021); Wang et al. (2022); Liu et al. (2022a); Park et al. (2022). However, most existing works operate under the static paradigm where either the queries or the feature rep-

resentations are fixed during the detection process. For example, DETR3D Wang et al. (2022) and PETR series Liu et al. (2022a;b) use static queries for 3D object detection. Such static methods often overlook the complex spatial-temporal dynamics present in real-world data. Our work fundamentally differs by introducing a dynamic paradigm where queries are capable of iterative adaptation, thus effectively capturing intricate relationships in both spatial and temporal dimensions.

### 2.4 Temporal Information in Object Detection

Incorporating temporal information has been explored in various works Li et al. (2022c); Park et al. (2022); Liu et al. (2023). However, these methods often introduce significant computational complexity and are constrained by the static nature of their query or feature representations. Our Lightweight Temporal Fusion Module (LTFM) not only efficiently integrates temporal context but does so in a dynamic manner, further emphasizing the shift towards a dynamic paradigm in 3D object detection.

## 3 Method

In this section, we introduce DynamicBEV, a novel method designed for effective and efficient 3D object detection. Traditional static query-based methods lack the dynamism required to capture the diverse nature of 3D spaces. In contrast, DynamicBEV harnesses dynamic queries that undergo iterative updates, and thereby achieves unparalleled adaptability in discerning diverse object attributes. The key components of DynamicBEV is illustrated in Figure 1, and a comparison between static and dynamic query-based methods is shown in Figure 2.

DynamicBEV is composed of multiple integral components that synergize to facilitate robust and precise 3D object detection. The framework includes a backbone network responsible for initial feature extraction. With the extracted feature, a Dynamic Query Evolution Module (DQEM) comes into play. First, DQEM exploits K-means clustering to groups features around each query, which brings adaptive structure representation for complex 3D scenarios. Afterwards, a Top-K Attention module is employed by DQEM to iteratively refine queries with their associated feature clusters. Finally, a Lightweight Temporal Fusion Module (LTFM) is incorporated to efficiently capture temporal context for each query.

### 3.1 Dynamic Query Evolution Module (DQEM)

#### 3.1.1 Initialization of Queries (Pillars)

In the context of 3D object detection, the initialization of queries plays a pivotal role in the subsequent detection performance. In the BEV space, these queries, often referred to as "pillars", serve as reference points or anchors that guide the detection process. The query set $Q$ can be represented as:

$$Q = \{(x_i, y_i, z_i, w_i, l_i, h_i, \theta_i, v_{x_i}, v_{y_i})\}$$

where $(x_i, y_i, z_i)$ is the spatial coordinates of the $i$-th pillar, indicating its position in the BEV space. $w_i, l_i, h_i$ are width, length and height of the pillar, respectively, providing the shape attributes. $\theta_i$ is the orientation angle of the pillar, offering insights into its alignment in the BEV space. $v_{x_i}$ and $v_{y_i}$ are velocity components of the pillar, capturing its motion dynamics.

In traditional methods like SparseBEV Liu et al. (2023), these queries and their associate features are initialized based on pre-defined grid structures and remain static throughout the detection process. Such static nature are designed to capture general object patterns but is not adept at handling diverse scenarios with complex intricate object details. On the contrary, in DynamicBEV, the associated feature are grouped into a clustered structure, which well adapts to the complex 3D scene, and each pillar iteratively adjusts its attributes (like position, dimensions, or orientation) based on the associated feature clusters. Such dynamism renders the pillars better adaptability to the object attributes in the 3D scenes, leading to a more accurate and robust detection.

#### 3.1.2 K-means Clustering

In DynamicBEV, K-means clustering is first employed to divide the surrounding features $F$ of each query into $K$ clusters $C_1, \ldots, C_K$. The rationale behind employing K-means clustering lies in its ability to partition the feature space into clusters within which the feature variance is minimized. This enable each query to focus on groups of coherent features rather than unorganized points,

which is a more adaptive and structured representation, thereby enhancing the model's ability to discern the objects in 3D scenes. After K-means clustering, each query $q$ will have an associated set of feature clusters $C_k$, formally:

$$C_k = \{f_i \mid c_i = k\},$$

and the cluster center:

$$\mu_k = \frac{1}{|C_k|} \sum_{f_i \in C_k} f_i.$$

These clusters encapsulate the local patterns around each query, and provide the model with a more adaptive structured representation of the dynamic 3D scenes, serving as the foundation for the subsequent Top-K Attention steps.

### 3.1.3 TOP-K ATTENTION AGGREGATION

To allow each query to aggregate features in a dynamic way, we introduce a Top-K Attention mechanism. For each query $q$, we compute the attention weights over its associated feature clusters $C_k$ obtained from K-means clustering.

**Compute Attention Scores:** For each query feature $q$ and each cluster $C_k$, compute an attention score.

$$A_k = (W_q q)^T \cdot W_k \mu_k$$

Here, $W_q$ represents the weight vector for the query and $W_k$ represents the weight vector for the cluster. The dot product measures the relevance between the query and each cluster.

This step allows the model to measure the importance of each feature cluster with respect to the query, enabling more informed aggregations.

**Select Top-K Clusters:** Sort the attention scores $A_k$ in descending order and select the top-K clusters.

$$\text{Top-K clusters} = \text{argmax}_k(A_k), \quad k = 1, \dots, K$$

This selective attention mechanism enables each query to focus on the most relevant clusters, which may even be farther away, thus enriching the aggregated feature.

**Weighted Feature Aggregation:** Aggregate the selected clusters using their attention weights to form the aggregated feature $q'$ to update each query $q$.

$$q' = \sum_{k \in \text{Top-K}} \text{Softmax}(A)_k \cdot \mu_k$$

The weighted sum allows for a rich combination of features, enabling each query to adaptively focus on different aspects of the surrounding features.

The aggregated feature $q'$ serves as the foundation for 3D object prediction. By allowing each query to aggregate information even from distant clusters, the model's capacity to capture long-range dependencies is significantly enhanced. Such capacity is particularly crucial in 3D object detection, where objects might have parts that are spatially separated but are contextually related.

### 3.1.4 DIVERSITY LOSS FOR BALANCED FEATURE AGGREGATION

The proposed Top-K Attention mechanisms has the risk of focusing excessively on the most relevant features corresponding to each query. While this approach is effective in capturing dominant patterns, it often neglects the long-tail or less prominent features that could be critical for certain edge cases or specific scenarios. For example, in a 3D object detection task involving vehicles and pedestrians, focusing solely on the most relevant features might capture the overall shape of a vehicle but miss out on smaller but important details like side mirrors or indicators, which are essential for precise localization and classification.

To address this limitation, we introduce a Diversity Loss $L_{\text{div}}$. This loss function aims to balance the attention mechanism by ensuring that not only the most relevant but also the less prominent features are considered. Unlike conventional entropy-based losses, which are agnostic to the task at hand,

our Diversity Loss is meticulously crafted for 3D object detection, ensuring a balanced attention distribution across different feature clusters, formally:

$$L_{\text{div}} = -\sum_{k=1}^{K} p_k \log p_k,$$

where the following function serves as a critical component for stabilizing the gradient flow during the back-propagation process, especially when dealing with clusters of varying relevance:

$$p(k) = \frac{\exp(A_k)}{\sum_{j=1}^{K} \exp(A_j)}.$$

This Diversity Loss brings several advantages. Firstly, it promotes a balanced feature representation by encouraging the model to pay attention to a variety of features, not just the most prominent ones. This is particularly useful for capturing less obvious but potentially crucial features. Secondly, the approach enhances the model's robustness, allowing it to adapt better to different scenarios and noise levels. Lastly, it fosters a more comprehensive understanding of the data, thereby improving the model's generalization capabilities.

### 3.1.5 DYNAMIC ADAPTATION OF QUERIES

After initializing the queries as pillars and performing K-means clustering to obtain feature clusters $C_k$, the next crucial step is dynamically adapting these queries based on the Top-K Attention mechanism. This dynamic adaptation is the key difference from SparseBEV, where the queries are static. In DynamicBEV, each query not only captures the local information but also dynamically updates itself to aggregate relevant features from a large scope of feature clusters.

**Initial Feature Aggregation:** For each query $q$, aggregate the initial set of features using a simple average or any other aggregation method.

$$q \leftarrow \frac{1}{|F|} \sum_{f \in F} f$$

This initial aggregation serves as a baseline, capturing the immediate vicinity of the query. It acts as an anchor, grounding the subsequent dynamic adaptations.

**Top-K Attention Update:** Apply the previously described Top-K Attention mechanism to adaptively update each query $q$ using its associated feature clusters $C_k$.

$$q \leftarrow q' + \beta \cdot q$$

Here, $q'$ is the aggregated feature obtained from Top-K Attention, and $\beta$ is a hyper-parameter that controls the blending of initial and dynamically aggregated features.

This step allows each query to adaptively refine its feature representation based on both local and long-range information, enhancing its ability to capture complex patterns and relationships.

**Iterative Update:** Repeat the K-means clustering and Top-K Attention steps, using the newly updated queries $q$ as the new pillars for the next iteration. Such iterative update ensures the queries continuously adapting to the varying feature landscape, thereby increasing the model's robustness and adaptability.

By iteratively updating queries through a combination of K-means clustering and Top-K Attention, DynamicBEV ensures each query is both locally and globally informed, thereby capturing richer and more balanced feature representations. This dynamic adaptation is a significant advancement over SparseBEV, where pillars remain static and cannot adapt to capture long-range dependencies.

### 3.2 LIGHTWEIGHT TEMPORAL FUSION MODULE

In DynamicBEV, the key advantage of our Lightweight Temporal Fusion Module (LTFM) lies in its computational efficiency. Unlike traditional temporal fusion methods that rely on resource-intensive recurrent or convolutional layers, LTFM leverages the already computed dynamic queries $Q$ and their corresponding feature clusters $C_k$, thereby avoiding additional heavy computations.

**Temporal Query Initialization:** The temporal queries $q$ are initialized using a weighted combination of current and previous dynamic queries , thus reusing existing computations.

$$q \leftarrow \alpha \cdot q + (1 - \alpha) \cdot q_{\text{previous}}$$

By reusing the dynamic queries, we eliminate the need for separate temporal query extraction, thereby reducing computational overhead.

**Dynamic Temporal Aggregation:** The Top-K Attention mechanism is applied directly to $q$, reusing the previously computed feature clusters $C_k$ for both current and previous time steps.

$$q' = \text{Top-K Attention}(q, F_{\text{current}}, F_{\text{previous}})$$

This obviates the need for separate temporal feature extraction, further reducing computational cost.

**Query Update:** The temporal queries $q$ are updated using the aggregated temporal features $q'$, similar to the dynamic query update in the previous sections.

$$q \leftarrow q' + \beta \cdot q$$

The update operation is computationally light, as it only involves basic arithmetic operations, thus bringing the computational efficiency.

LTFM provides an efficient way to incorporate temporal context without introducing a significant computational burden. By reusing existing computations to avoid additional complex operations, LTFM offers a lightweight yet effective solution for temporal fusion.

### 3.3 COMPUTATIONAL COMPLEXITY

The computational efficiency of DynamicBEV is one of its key advantages. Below, we quantify this in terms of time complexity: The overall time complexity is approximately $O(nKId + n\log n + n)$, where $n$ is the number of data points, $K$ is the number of cluster centers, $I$ is the number of iterations in K-means, $d$ is the dimensionality of each data point. This is relatively low compared to methods that require more complex temporal fusion techniques such as RNNs or CNNs.

## 4 EXPERIMENT

### 4.1 IMPLEMENTATION DETAILS

We adopt ResNet He et al. (2016) as the backbone, the temporal module in our model is designed to be lightweight and we use a total of $T = 8$ frames by default, with an interval of approximately 0.5s between adjacent frames. For label assignment between ground-truth objects and predictions, we use the Hungarian algorithmKuhn (1955). The loss functions employed are focal lossLin et al. (2017) for classification and L1 loss for 3D bounding box regression, augmented by our custom Diversity Loss $L_{\text{div}}$ with a weight factor of $\lambda = 0.1$. The initial learning rate is $2 \times 10^{-4}$, and it is decayed using a cosine annealing policy. In line with recent advancements, we adjust the loss weight of $x$ and $y$ in the regression loss to 2.0, leaving the others at 1.0, to better capture spatial intricacies. We also incorporate Query Denoising to stabilize training and speed up convergence, as suggested by the recent work StreamPETRWang et al. (2023). For our K-means clustering, $K$ is set to 6. The number of Top-K clusters for attention is set to 4. The hyperparameter $\beta$ used for blending in query update is set to 0.6, and $\alpha$ for temporal fusion in the Lightweight Temporal Fusion Module (LTFM) is set to 0.4.

### 4.2 DATASETS AND EVALUATION CRITERIA

Our experiments utilize the nuScenes dataset Caesar et al. (2020), a rich source of multimodal sensor information encompassing 1000 driving sequences, each lasting around 20 seconds. Annotations are available at a rate of 2Hz for key frames. Each frame in the dataset offers a comprehensive 360-degree field of view through six camera sensors. For the task of 3D object detection, the dataset incorporates approximately 1.4 million 3D bounding boxes across 10 categories of objects.

We adopt a similar task setting as in previous worksLiu et al. (2023) for Birds-Eye View (BEV) segmentation. The official evaluation metrics of nuScenes are comprehensive; they not only include mean Average Precision (mAP), which is calculated based on the center distance in the

| Method | Backbone | Input Size | Epochs | NDS | mAP | mATE | mASE | mAOE | mAVE | mAAE |
|--------|----------|-----------|--------|-----|-----|------|------|------|------|------|
| PETRv2 Liu et al. (2022b) | ResNet50 | $704 \times 256$ | 60 | 45.6 | 34.9 | 0.700 | 0.275 | 0.580 | 0.437 | 0.187 |
| BEVStereo Li et al. (2022a) | ResNet50 | $704 \times 256$ | 90 | 50.0 | 37.2 | 0.598 | 0.270 | 0.438 | 0.367 | 0.190 |
| BEVPoolv2 Huang & Huang (2022b) | ResNet50 | $704 \times 256$ | 90 | 52.6 | 40.6 | 0.572 | 0.275 | 0.463 | 0.275 | 0.188 |
| SOLOFusion Park et al. (2022) | ResNet50 | $704 \times 256$ | 90 | 53.4 | 42.7 | 0.567 | 0.274 | 0.511 | 0.252 | 0.181 |
| Sparse4Dv2 Lin et al. (2023) | ResNet50 | $704 \times 256$ | 100 | 53.9 | 43.9 | 0.598 | 0.270 | 0.475 | 0.282 | 0.179 |
| StreamPETR † Wang et al. (2023) | ResNet50 | $704 \times 256$ | 60 | 55.0 | 45.0 | 0.613 | 0.267 | 0.413 | 0.265 | 0.196 |
| SparseBEV Liu et al. (2023) | ResNet50 | $704 \times 256$ | 36 | 54.5 | 43.2 | 0.619 | 0.283 | 0.396 | 0.264 | 0.194 |
| SparseBEV †Liu et al. (2023) | ResNet50 | $704 \times 256$ | 36 | 55.8 | 44.8 | 0.595 | 0.275 | 0.385 | 0.253 | 0.187 |
| **DynamicBEV** | ResNet50 | $704 \times 256$ | 60 | 55.9 | 45.1 | 0.606 | 0.274 | 0.387 | 0.251 | 0.186 |
| **DynamicBEV** † | ResNet50 | $704 \times 256$ | 60 | **57.0** | **46.4** | 0.581 | 0.271 | 0.373 | 0.247 | 0.190 |
| DETR3D † Wang et al. (2022) | ResNet101 | $1600 \times 900$ | 24 | 43.4 | 34.9 | 0.716 | 0.268 | 0.379 | 0.842 | 0.200 |
| BEVFormer † Li et al. (2022c) | ResNet101 | $1600 \times 900$ | 24 | 51.7 | 41.6 | 0.673 | 0.274 | 0.372 | 0.394 | 0.198 |
| BEVDepth Li et al. (2022b) | ResNet101 | $1408 \times 512$ | 90 | 53.5 | 41.2 | 0.565 | 0.266 | 0.358 | 0.331 | 0.190 |
| Sparse4D † Lin et al. (2022) | ResNet101 | $1600 \times 900$ | 48 | 55.0 | 44.4 | 0.603 | 0.276 | 0.360 | 0.309 | 0.178 |
| SOLOFusion Park et al. (2022) | ResNet101 | $1408 \times 512$ | 90 | 58.2 | 48.3 | 0.503 | 0.264 | 0.381 | 0.246 | 0.207 |
| SparseBEV † Liu et al. (2023) | ResNet101 | $1408 \times 512$ | 24 | 59.2 | 50.1 | 0.562 | 0.265 | 0.321 | 0.243 | 0.195 |
| **DynamicBEV** † | ResNet101 | $1408 \times 512$ | 24 | **60.5** | **51.2** | 0.575 | 0.270 | 0.353 | 0.236 | 0.198 |

Table 1: Performance comparison on nuScenes `val` split. † benefits from perspective pretraining.

ground plane instead of 3D IoU, but also feature five additional True Positive (TP) error metrics: ATE, ASE, AOE, AVE, and AAE, to measure the errors in translation, scale, orientation, velocity, and attributes respectively. To provide a unified score that captures multiple facets of detection performance, the nuScenes Detection Score (NDS) is used, defined as $NDS = \frac{1}{10} \left[ 5 \times mAP + \sum_{mTP \in TP} (1 - \min(1, mTP)) \right]$.

## 4.3 COMPARISON WITH THE STATE-OF-THE-ART METHODS

Table 1 presents the performance of our DynamicBEV compared with other state-of-the-art methods, which outperforms all other methods by a considerable margin on the nuScenes validation dataset. With a ResNet50 backbone and an input size of $704 \times 256$, DynamicBEV achieves a nuScenes Detection Score (NDS) of $55.9$, which is marginally higher than the $54.5$ achieved by SparseBEV. More significantly, when perspective pre-training is applied, indicated by the † symbol, the NDS score of DynamicBEV rises to $57.0$, outperforming the $55.8$ by SparseBEV.

In more complex configurations, such as using a ResNet101 backbone and an input size of $1408 \times 512$, DynamicBEV outshines its competitors with an NDS of $60.5$, exceeding SparseBEV's $59.2$, making it the current leading approach.

DynamicBEV consistently maintains high Mean Average Precision (mAP) scores, proving its robust object detection capabilities. In terms of True Positive metrics like mATE, mASE, DynamicBEV holds its ground well compared to SparseBEV and other competing methods. Moreover, the model also performs well on fine-grained evaluation metrics such as Object Orientation Error (mAOE) and Attribute Error (mAAE). The application of perspective pre-training not only improves nearly all evaluation metrics but also showcases the model's adaptability and flexibility.

The advantages of DynamicBEV primarily stem from two inherent aspects: Firstly, the design of DynamicBEV allows it to better capture long-range dependencies. In 3D object detection, different parts of an object might be spatially distant but contextually related. For instance, the front and rear of a car might be far apart in the BEV space, yet they belong to the same object. SparseBEV, being a static query-based method, might struggle in such scenarios since its query points are fixed and cannot dynamically adapt to the changing scene. In contrast, DynamicBEV, through its Dynamic Query Evolution Module, can update its query points in real-time, thereby better capturing these long-range dependencies. Secondly, DynamicBEV is better equipped to handle the dynamism of real-world scenes. Objects in real-world scenarios might move, rotate, or change their shape. SparseBEV, with its static query points, might falter in such dynamically changing scenes. However, DynamicBEV, through its dynamic queries and K-means clustering, can dynamically adjust its query points, thus better adapting to the evolving scene. In the following section, we will further validate these observations through ablation experiments.

## 4.4 ABLATION STUDY

### 4.4.1 DYNAMIC QUERY EVOLUTION MODULE (DQEM)

For all ablation studies, we use ResNet-50 as the backbone and adopt the same training and evaluation protocols. The baseline model employs the standard cross-attention mechanism. The Dynamic-

K Block integrates Dynamic Queries, K-means Clustering, and Top-K Attention as a unified module. We compare this with the baseline model that uses standard cross-attention.

| Model Configuration | NDS | mAP |
|---|---|---|
| Baseline (Cross-Attention) | 51.7 | 40.8 |
| Dynamic-K Block | 55.9 | 45.1 |

Table 2: Ablation study on the Dynamic-K Block.

| Model Configuration | NDS | mAP |
|---|---|---|
| Baseline (No Temporal Fusion) | 52.8 | 42.3 |
| With LTFM | 55.9 | 45.1 |
| LSTM-based Fusion | 53.5 | 43.2 |
| Convolutional LSTM Fusion | 53.7 | 43.5 |
| Simple Averaging | 52.5 | 42.0 |

Table 3: Ablation study on the Lightweight Temporal Fusion Module (LTFM).

Table 2 shows that the introduction of the Dynamic-K Block results in an 4.2% increase in NDS and a 4.3% increase in mAP compared to the baseline. The Dynamic-K Block's significant performance boost can be attributed to its ability to focus on key features dynamically. Traditional methods with static query points, like the baseline model, might not be able to adapt to the dynamic nature of real-world scenes. In contrast, the Dynamic-K Block, with its integration of Dynamic Queries, K-means Clustering, and Top-K Attention, allows the model to dynamically adjust its focus based on the scene's context. This adaptability ensures that the model can give precedence to critical features, especially in complex scenes where objects might be occluded or distant from each other.

To further understand the impact of the clustering mechanism on the performance of DynamicBEV, we explored alternative clustering methods in Table 4. Specifically, we evaluated the performance of DBSCAN and Agglomerative Hierarchical Clustering, comparing them with our default choice, K-means. From the results, K-means notably surpasses DBSCAN and Agglomerative Hierarchical Clustering in NDS and mAP. K-means' consistent partitioning aligns with 3D object detection's dynamic nature, ensuring coherent feature focus. Its computational efficiency is vital for large-scale tasks, unlike the less scalable Agglomerative method. Unlike density-dependent DBSCAN, K-means' density independence ensures adaptability across varied scenarios. The clear centroid representation in K-means enhances the subsequent Top-K Attention step.

| Model Configuration | NDS | mAP |
|---|---|---|
| K-means | 55.9 | 45.1 |
| DBSCAN | 52.3 | 41.8 |
| Agglomerative | 53.1 | 42.5 |

Table 4: Impact of the clustering mechanism on the performance of DynamicBEV.

| Temporal Resolution | NDS | mAP |
|---|---|---|
| Every Frame | 55.5 | 44.8 |
| Every 2 Frames | 55.9 | 45.1 |
| Every 5 Frames | 55.2 | 44.5 |

Table 5: Performance of LTFM at different temporal resolutions.

| Diversity Loss | NDS | mAP |
|---|---|---|
| Without | 54.4 | 43.7 |
| With | 55.9 | 45.1 |

Table 6: Impact of Diversity Loss in feature aggregation.

### 4.4.2 LIGHTWEIGHT TEMPORAL FUSION MODULE (LTFM)

To study the effectiveness of our Lightweight Temporal Fusion Module (LTFM), we compare it with the baseline that doesn't employ temporal fusion and other prevalent temporal fusion methods in Table 3. All other configurations remain the same for a fair comparison.

Incorporating the Lightweight Temporal Fusion Module (LTFM) to the baseline model results in a 3.1% increase in NDS and a 2.8% increase in mAP. These improvements indicate that LTFM effectively captures the temporal dependencies without introducing significant computational overhead, thus validating its utility in our DynamicBEV framework. The LTFM provides the model with crucial context about these object movements. By fusing information across time, the model gains a more comprehensive understanding of the scene, allowing it to predict object trajectories and interactions more accurately. LTFM consistently outperformed other methods like LSTM-based fusion, Convolutional LSTM fusion, and simple averaging across time. This can be attributed to LTFM's lightweight design and its adeptness at capturing crucial temporal dependencies without significant computational overhead.

We further explored the temporal resolution at which the LTFM operates in Table 5. Different scenarios might benefit from different temporal granularities. When comparing the performance of LTFM at different time intervals, such as every frame, every 2 frames, and every 5 frames, we observed that fusing information at every 2 frames provided the optimal balance between computational efficiency and detection accuracy.

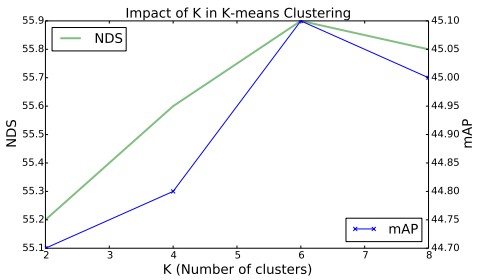 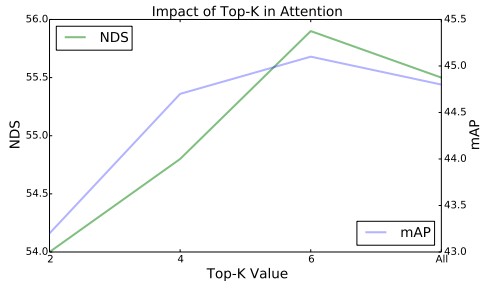

(a) Different number of clusters $K$ in K-means.      (b) Different Top-K values in Top-K Attention.

Figure 3: Performance impact of different parameter settings in K-means and Top-K Attention.

### 4.4.3 SELECTION OF $K$ IN K-MEANS AND TOP-K ATTENTION

As illustrated in Figure 3a, increasing the number of clusters $K$ initially improves both NDS and mAP. The performance plateau observed after $K = 6$ in K-means clustering suggests that there's an optimal number of clusters that capture the scene's essence. Having too many clusters might over-segment the data, leading to redundant or even conflicting information. Similarly, Figure 3b shows that utilizing Top-K Attention with $K = 6$ yields the best performance, highlighting the importance of selective attention. Including Diversity Loss improves both NDS and mAP, as shown in Table 6, indicating its effectiveness in balancing the attention mechanism and capturing a variety of features.

### 4.4.4 PARAMETER SENSITIVITY IN DYNAMIC ADAPTATION AND TEMPORAL FUSION

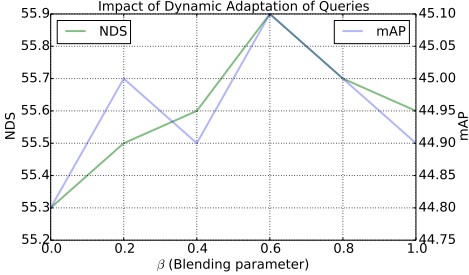 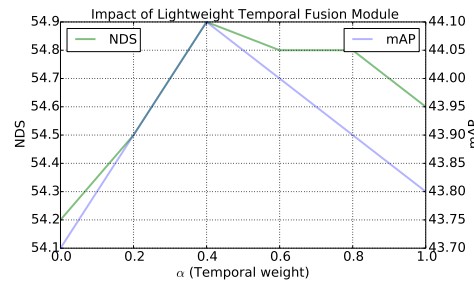

(a) $\beta$ in Dynamic Adaptation of Queries.      (b) $\alpha$ in Lightweight Temporal Fusion Module.

Figure 4: Sensitivity analysis of parameters $\beta$ and $\alpha$ in the model.

The optimal values for key parameters are discussed with respect to their impact on model performance. As shown in Figure 4a, the optimal value for $\beta$ is around 0.6, providing the best blend of initial and dynamically aggregated features. Deviating too much from this value results in suboptimal performance. Similarly, Figure 4b shows that the value of $\alpha = 0.4$ yields the highest NDS and mAP, suggesting that balancing the current and previous dynamic queries effectively captures temporal information.

## 5 CONCLUSION

In this paper, we presented DynamicBEV, a novel approach to 3D object detection that leverages dynamic queries in BEV space. Distinct from conventional static query-based techniques, DynamicBEV iteratively adapts queries to capture complex spatial and temporal relationships within the data. This dynamic paradigm offers a more flexible and adaptive mechanism for 3D object detection, effectively constituting a new frontier in the field.

Our method integrates various novel components, including K-means clustering for feature selection, Top-K Attention for adaptive feature aggregation, and a Lightweight Temporal Fusion Module for efficient temporal context integration. These components collectively enable our model to outperform state-of-the-art methods on various benchmarks, thus validating the efficacy of the dynamic query-based paradigm.

As future work, we aim to explore the applicability of dynamic queries in other vision tasks and to further optimize the computational efficiency of our model. We also plan to investigate the potential of incorporating more advanced temporal models to capture long-term dependencies in videos or large-scale 3D scenes.

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
