# OpenReview forum: "DynamicBEV: Leveraging Dynamic Queries and Temporal Context for 3D Object Detection"
_ICLR.cc/2024/Conference — ICLR 2024 Conference Withdrawn Submission_

### Official Review · Reviewer_Pvpq · 2023-10-30

**Soundness:** 3 good
**Presentation:** 2 fair
**Contribution:** 2 fair
**Rating:** 3
**Confidence:** 4

**Summary:**

This paper proposes a lightweight and effective method for aggregating BEV pillar features using K-means clustering and Top-K Attention. The authors also introduce a Diversity Loss to prevent the attention mechanism from focusing too heavily on the most relevant features. The proposed method is evaluated on the nuScenes dataset and outperforms previous methods.

**Strengths:**

The proposed clustering and top-K attention mechanism are simple and intuitive, yet achieve strong performance compared to previous state-of-the-art methods (Table 1). Extensive ablation studies in Section 4.4 demonstrate the benefits of the proposed modules.

**Weaknesses:**

Latency: Section 3.3 states that "the computational efficiency of DynamicBEV is one of its key advantages”. However, not all floating-point operations (FLOPs) are created equal, especially for the clustering operation on GPUs, TPUs, and other edge devices. It would be helpful if the authors could measure the latency of the full model and provide a breakdown of the latency of each component (e.g., clustering, sorting top-k).

Generalization: Evaluating the proposed method on only one dataset is not sufficient. I suggest evaluating the proposed method on at least one additional dataset.

Visualization: Could the authors provide detailed illustrations on K-mean clustering and Top-K attention in Fig1?
Fig 2 is not clear.  What does each color mean?

**Questions:**

Please see weaknesses.

---

### Official Review · Reviewer_hs8x · 2023-10-31

**Soundness:** 2 fair
**Presentation:** 2 fair
**Contribution:** 2 fair
**Rating:** 5
**Confidence:** 3

**Summary:**

The authors explore and analyze the existing query-based paradigm for 3D BEV-based object detection, and propose to adopt dynamic queries to do temporal feature extraction. The experimental results on nuScenes, show the effectiveness of the proposed method.

**Strengths:**

The task of query-based paradigm for 3D BEV-based object detection is popular and interesting in the 3D community. The authors propose to use dynamic queries to do feature learning and make it work on nuScenes.

**Weaknesses:**

1. Performance difference on large-scale vs small-scale objects. It would be interesting if the authors could show the detailed detection performance of 10 classes on the nuScenes. From my understanding, the proposed method is kind of sensitive to different objects with different sizes.

2. It is unclear to me how to define the size of associated feature cluster, and the number of the query.

3. More quantitative/qualitative results. The manuscript does report detection numbers on nuScenes validation set, however, the authors forgot to compare their methods with recent SOTAs on the test set. Also, it would be much convincing if the authors can present some qualitative results or report the results on more public datasets, i.e., KITTI or Waymo.

4. I am curious about the inference time of the proposed method. The authors repeatedly claimed that the traditional temporal fusion is heavy computation, however, the attention computation is also heavy from my understanding.

**Questions:**

Please refer to the weakness part.

---

### Official Review · Reviewer_P6ky · 2023-10-31

**Soundness:** 2 fair
**Presentation:** 3 good
**Contribution:** 2 fair
**Rating:** 5
**Confidence:** 4

**Summary:**

This paper presents dynamic queries for 3D object detection in bird's-eye view, distinguishing it from the static queries employed in SparseBEV. To enhance the model's performance, the authors have introduced K-means clustering and Top-K Attention mechanisms, which facilitate the integration of global features into the queries. Additionally, the paper introduces a diversity loss to encourage queries to focus on all clustered features. Then, a Lightweight Temporal Fusion Module is illustrated to speed up multi-frame fusion by using pre-computed features.

**Strengths:**

1. I would like to compliment you on the clear and concise language used throughout the manuscript, as well as the well-designed figures and tables. These elements greatly enhance the readability and understandability of the paper.
2. It is clever to use clustering attention to reduce the computation cost of global attention.

**Weaknesses:**

1. The proposed dynamic queries is not new. Prior research, such as CMT[1] and UVTR[2], has already demonstrated the adjustment of queries in each decoder layer. Moreover, CMT employs global attention, while UVTR utilizes local attention to update query features, raising questions about the novelty of the proposed dynamic queries.

2. There are some experiment omissions that limit the comprehensiveness of the evaluation. Notably, there is a lack of crucial comparisons, such as latency comparisons with SparseBEV and an analysis of the performance-to-latency trade-off when employing clustering attention in contrast to the global attention mechanism used in CMT.

[1] Cross Modal Transformer: Towards Fast and Robust 3D Object Detection, in ICCV 2023.
[2] Unifying Voxel-based Representation with Transformer for 3D Object Detection, in NeurIPS 2022.

**Questions:**

Please see the Weaknesses.

---

### Official Review · Reviewer_NE19 · 2023-11-03

**Soundness:** 2 fair
**Presentation:** 2 fair
**Contribution:** 2 fair
**Rating:** 5
**Confidence:** 4

**Summary:**

The authors introduce a new paradigm in DynamicBEV, a novel approach that employs dynamic queries for BEV-based 3D object detection. The proposed dynamic queries exploit K-means clustering and Top-K Attention creatively to aggregate information more effectively from both local and distant features, which enables DynamicBEV to adapt iteratively to complex scenes. LTFM is designed for efficient temporal context integration with a significant computation reduction.

**Strengths:**

1. k-means clustering determines how pillars fit into localized patterns and features in 3D space, facilitating a detailed understanding of the characteristics of the object.
2. Diversity loss ensuring that the model is not overly focused on dominant features promotes a balanced focus on the clustering of various features
3. LTFM embodies the essence of computational efficiency and relieves the need for resource-intensive operations by leveraging existing calculations to manage temporal context
4. DynamicBEV outperforms sota methods on the nuScenes validation dataset

**Weaknesses:**

1. Missing nuScenes test results and the paper is difficult to understand and lacks the necessary visualizations
2. What do the surrounding features mean, and can authors explain dividing the surrounding features F of each query into K clusters C1, . . . , CK? Why use k-means to cluster features? And Fig.3 (a) does not show a big gap between k=5,6 or 7. What is aggregate based on? Is it the distance between features?
3. Why use tok-k attention? If the authors want local information, deformable attention is a choice.
4. The authors use Iterative Update and repeat the K-means clustering and Top-K Attention steps. So, I think the authors should report the inference speed.
5. For LTFM, authors should compare with StreamPETR for a fair comparison.

Minor: Fig.2 shows the difference between static query and dynamic query-based methods but lacks detailed explanations. (Similar in Fig.1)

**Questions:**

see weakness